# Fabrication of Novel CeO$_2$/GO/CNTs Ternary Nanocomposites with Enhanced Tribological Performance

**Chunying Min** [1,2,3,4,*], **Zengbao He** [1,2], **Haojie Song** [5], **Dengdeng Liu** [1,2], **Wei Jia** [1,2], **Jiamin Qian** [1,2], **Yuhui Jin** [1,2] **and Li Guo** [1,2,*]

1   Research School of Polymeric Materials, Jiangsu University, Zhenjiang 212013, China;
    2221705033@stmail.ujs.edu.cn (Z.H.); 2211605028@stmail.ujs.edu.cn (D.L.);
    2211805067@stmail.ujs.edu.cn (W.J.); 3160705035@stmail.ujs.edu.cn (J.Q.);
    3170705065@stmail.ujs.edu.cn (Y.J.)
2   School of Material Science & Engineering, Jiangsu University, Zhenjiang 212013, China
3   State Key Laboratory of Tribology, Tsinghua University, Beijing 100084, China
4   National United Engineering Laboratory for Advanced Bearing Tribology, Henan University of Science and
    Technology, Luoyang 471023, China
5   School of Materials Science & Engineering, Shaanxi University of Science & Technology, Xi'an 710021, China;
    songhaojie@sust.edu.cn
*   Correspondence: mj790206@ujs.edu.cn (C.M.); liguo@ujs.edu.cn (L.G.); Tel.: +86-0511-8878-0190 (C.M.)

**Abstract:** Increasing demands of multi-functional lubricant materials with well distributed nanoparticles has been generated in the field of oil lubrication. In this study, one-dimensional (1-D) acidified multi-walled carbon nanotubes (CNTs) and two-dimensional (2-D) graphene oxide (GO) sheets were dispersed together under an ultra-sonication condition to form CNTs/GO hybrids and the corresponding CNTs/GO hybrids decorated with uniform zero-dimensional (0-D) cerium oxide (CeO$_2$) nanoparticles were prepared via a facile hydrothermal method. The tribological performance of CeO$_2$/CNTs/GO ternary nanocomposite was systematically investigated using a MS-T3000 ball-on-disk tester. The results demonstrated that CeO$_2$/GO/CNTs nanocomposites can effectively reduce the friction of sliding pairs in paraffin oil. Moreover, the oil with 1 wt% of CeO$_2$/GO/CNTs exhibited the best lubrication properties with the lowest friction coefficient and wear scar diameters (WSD) compared with adding only GO nanosheet, CeO$_2$, and CeO$_2$/CNTs hybrid nanocomposite as lubricant additives. It is concluded that due to the synergistic effect of 0D CeO$_2$, 1D CNTs, and 2D GO during sliding process, a dimensionally mixed CeO$_2$/GO/CNTs nanocomposite exhibits excellent lubricating properties, providing innovative and effective additives for application in the field of lubrication.

**Keywords:** multi-walled carbon nanotubes; graphene oxide; cerium oxide; lubricating oil additives

## 1. Introduction

In the continuous development and research of tribology and nanomaterials, except for the improvement of traditional lubricant additives, many researchers focus on introducing nanomaterials into tribology for exploration [1–7]. Nanomaterials exhibit better anti-friction and wear resistance performance under more severe conditions, which makes them well-studied and applied in terms of friction [8,9]. Nanomaterials generally have a large specific surface area and surface tension, with the size between 1 and 100 nm. After the surface is modified, it can be uniformly and stably dispersed in some organic solvents [10–13]. The uniform dispersion of nanomaterials in the base oil

can effectively improve its friction and wear resistance and possess excellent properties that other traditional lubricating oils do not have.

Cerium (Ce), a zero-dimensional (0D) nanoparticle, is a kind of rare earth element. Compared with metal nanoparticles, Ce is difficult to oxidize and exhibits excellent tribological properties in marine, aerospace, and other environments. Ce and its compounds as filler can effectively improve the tribological properties of the materials. Zhuo et al. [14] used $CeO_2$ and Cu nanoparticles as lubricating oil additives, and modified $CeO_2$ and Cu nanoparticles using a suitable surfactant mixture to achieve good dispersion and stability in the lubricating oil.

The special size, unique one-dimensional (1-D) tubular hollow and topology structure of carbon nanotubes make them attractive for various applications, including fabricating nanotubes reinforced nanocomposites on mechanical properties and tribological properties [15–17]. However, the insufficient dispersibility of carbon nanotubes deter their further research and development. The carbon nanotubes are acidified to have a partial oxygen-containing group on the surface to change their inertness. The modified carbon nanotubes are less likely to agglomerate and have increased solubility in solvent. Additionally, the modified carbon nanotubes are more easily combined with other materials during reaction. Gofman et al. [18] used carbon nanotubes containing a –COOH group as filling materials for polyimide, which showed significantly improved mechanical and tribological properties. Chen et al. [19] prepared carbon nanotubes using a mixture of sulfuric acid and nitric acid. The experimental results showed that the dispersion of the modified carbon nanotubes in base oil is improved, and the antifriction and wear resistance of the modified carbon nanotubes are also enhanced effectively.

Graphene oxide (GO), a two-dimensional (2-D) lattice, has numerous oxygen-containing groups and has superior thermal conductivity and mechanical properties. In addition, negative charge on the surface of GO makes it more convenient to combine with other materials without additional processing. As a new material, GO is widely used in various fields [20–22] such as tribology. Kim et al. [23] used an electrokinetic spraying process to coat graphene oxide on silicon wafer in order to use the coating as solid lubricant. Results showed that the coating has better surface protection ability and lower friction coefficient.

In recent years, many studies have confirmed that nano-hybrid materials can fully take advantage of various fillers and enhance the tribological properties of lubricating oil, polymers, and other materials [24–26]. Xin et al. [27] prepared a self-lubricating graphene oxide/nano-$MoS_2$ hybrid using a three-step method and discussed the synergistic effect of self-lubricating and anti-wear properties of the hybrid. Results show that the friction coefficient and wear rate of the base oil decreased by 25% and 64%, respectively, after the addition of nanocomposites. Zhang et al. [28] indicated that compared with pure liquid lubricant and liquid lubricant with $MoS_2$, the friction coefficient of liquid lubricant with CNTs/$MoS_2$ hybrids is decreased by 15%, due to the synergistic effect. Moreover, Bai et al. [29] studied a ceria/graphene composite as lubricating oil additive and considered the synergy between $CeO_2$ and graphene as an effective way to improve the tribological properties; however, the dispersion stability of lubricating additive and antifriction performance need further improvement.

In this work, we successfully prepared CNTs by using mixed acid. Additionally, different weight ratios of 0-D $CeO_2$ nanoparticle, 1-D CNTs, and 2-D GO were compounded to obtain various $CeO_2$/GO/CNTs nanocomposites via a hydrothermal method. CNTs and GO can form a stable structure through π-π bonding [30,31], and the surface is rich in oxygen-containing groups that can combine well with $CeO_2$. As far as we know, there are rare reports on the friction and wear properties of $CeO_2$/GO/CNTs hybrid as lubricating additive of base oil. Excellent structural properties of $CeO_2$/GO/CNTs nanocomposites can achieve synergistic effect among $CeO_2$, CNTs, and GO, which can provide some ideas for preparing high-performance oil lubricating additives.

## 2. Experimental

### 2.1. Materials

Cerium nitrate hexahydrate ($Ce(NO_3)_3 \cdot 6H_2O$), sodium hydroxide (NaOH), anhydrous ethanol were all purchased from Sinopharm Chemical Reagent Co., Ltd., Shanghai, China. Multi-walled carbon nanotubes (20–30 nm outer diameters, 10–30 $\mu$m length) were purchased from Chengdu organic chemistry Co., Ltd., Chengdu, China. GO was prepared using the sophisticated Hummers method [32] with natural flake graphite powders as raw material. Nitric acid ($HNO_3$) and sulfuric acid ($H_2SO_4$) were purchased from East Instrument Chemical Glass Co. Ltd., Zhenjiang, China.

### 2.2. Preparation of $CeO_2$/GO/CNTs Nanocomposites

Pure carbon nanotubes were treated with mixed acid of $H_2SO_4$ and $HNO_3$ (3:1) at 80 °C for 2 h under stirring, then washed and dried. The as-prepared CNTs and GO (total mass of GO and CNTs was 75 mg) were dispersed in a 37.5 mL mixed solution of ethanol and water using sonication to form the GO/CNTs uniform dispersion. Simultaneously, $Ce(NO_3)_3 \cdot 6H_2O$ was dispersed in a 37.5 mL mixed solution of ethanol and water, followed by pouring $Ce(NO_3)_3 \cdot 6H_2O$ solution into the GO/CNTs uniform dispersion with magnetic stirring for one hour. After that, the mixed solution was transferred into a Teflon-lined autoclave with addition of a certain amount of NaOH while gently stirring. The autoclave was thermal treated in an oven at 140 °C for 12 h. To obtain the $CeO_2$/GO/CNTs nanocomposites, the as-prepared products were washed by water and ethanol for five times, and freeze-drying for 24 h. $CeO_2$/CNTs nanocomposites was obtained following the above steps except that 75 mg GO/CNTs was substituted with 75 mg CNTs.

When the mass ratio of GO and CNTs was 3:1, GO/CNTs and $Ce(NO_3)_3 \cdot 6H_2O$ had different mass ratios (1:1, 3:1, 4:1) according to the same process to obtain various nanocomposites ($CeO_2$/GO/CNTs-a, $CeO_2$/GO/CNTs-b, $CeO_2$/GO/CNTs-c, respectively). When mass ratio of GO/CNTs and $Ce(NO_3)_3 \cdot 6H_2O$ was 3:1, GO/CNTs had different mass ratios (4:1, 5:1) according to the above steps to obtain various nanocomposites ($CeO_2$/GO/CNTs-d, $CeO_2$/GO/CNTs-e, respectively).

### 2.3. Characterization

The Raman measurements were performed on a Renishaw Microscope System RM2000 (Illinois, USA) with a 20 mW $Ar^+$ laser at 532 nm. The variation of elemental chemical state and surface chemical components were tested using X-ray photoelectron spectroscopy (XPS, ESCALAB 250XI, Thermo Fisher Scientific, Waltham, MA USA) with an Al K$\alpha$ source. The morphologies and structures of different $CeO_2$/GO/CNTs nanocomposites were characterized using transmission electron microscopy (TEM, JEM-2100, Philips, Netherlands).

### 2.4. Analysis of Tribological Performance

The tribological properties of the specimen were determined by the friction test on the MS-T3001 ball-on-disk machine (the rotation speed was 300 r/min, the load was 10 N, the radius of rotation was 3 mm, and the duration was 30 min). The balls used in the experiment was made of GCr15 bearing steel with a hardness of 62 HRC and a diameter of 4 mm. The steel disk was made of 45 steel and polished to a bright surface with different specifications of sandpaper before the friction test. The width of the wear scar is an important parameter for characterizing the friction performance and measured using a Zeiss Observer Z1m metalloscope (Oberkochen Deutschland). The friction test was repeated three times for each specimen.

## 3. Results and Discussion

The flow diagram of the preparation of $CeO_2$/GO/CNTs nanocomposites is shown in Figure 1. The CNTs and GO nanosheets can combine together by covalent bonds due to the existence of a

plurality of oxygen-containing groups on the surface of both CNTs and GO. In addition, $Ce^{3+}$ can also be adsorbed on the surface of GO and CNTs by covalent bonding. Subsequently, NaOH solution was added to the mixed solution to provide a large amount of free $OH^-$ which provided hydrolysis conditions for $Ce^{3+}$. The hydrothermal reaction is carried out to form $Ce(OH)^{2+}$, $Ce(OH)_2^+$ or $Ce(OH)_3$ compound, which were converted into $CeO_2$ via a hydrothermal reaction under high temperature and pressure. Then, in the reaction vessel, these products are converted into more stable $CeO_2$ with the high temperature and pressure hydrothermal reaction progresses, and finally the $CeO_2/GO/CNTs$ nanocomposite is obtained.

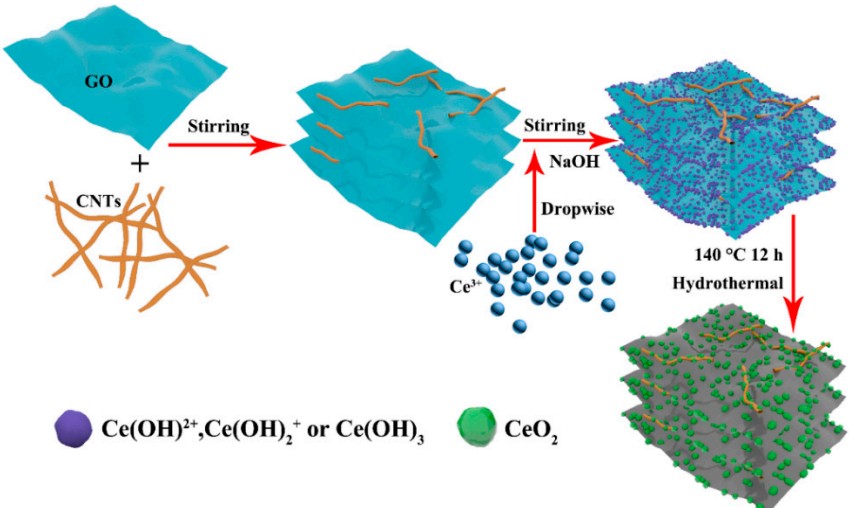

**Figure 1.** The formation processes of $CeO_2/GO/CNTs$ nanocomposites.

The morphologies of $CeO_2/CNTs$ and $CeO_2/GO/CNTs$ nanocomposites were characterized by TEM as shown in Figure 2. In Figure 2a, numerous $CeO_2$ nanoparticles were unevenly deposited on CNTs which were intertwined with each other accompanied by agglomeration. Figure 2b–f shows the microstructure of $CeO_2/GO/CNTs$ with different weight ratios of $CeO_2$, GO, and CNTs. From Figure 2b–f, it can be seen that GO exists as a carrier where CNTs combines and $CeO_2$ nanoparticles deposits. Specifically, via TEM observation of $CeO_2/GO/CNTs$-d nanocomposites (Figure 2e), CNTs and $CeO_2$ nanoparticles displayed excellent uniformity compared with other nanocomposites. It can be speculated that $CeO_2/GO/CNTs$-d nanocomposite is more suitable as lubricant additives in terms of the micromorphology of nanocomposites.

Raman spectroscopy is an efficient approach to prove the electronic structure of $CeO_2/GO/CNTs$ nanocomposites, especially for the changes of the C=C, C–C, and defects. Figure 3 shows the Raman spectra of GO, $CeO_2/CNTs$, and $CeO_2/GO/CNTs$ nanocomposites. From the Raman spectra of GO, there are characteristic bands related to the D ($\approx$1333 $cm^{-1}$) and G (1583 $cm^{-1}$) band in both GO and CNTs [33,34]. Apart from the D and G bands, the distinct strong peak at 456.13 $cm^{-1}$ corresponds to the characteristic peak of $CeO_2$ nanoparticles [35]. However, in the Raman spectra of the $CeO_2/GO/CNTs$ nanocomposite, there were only one set of characteristic peaks, which appeared at 1333.54 $cm^{-1}$ and 1583.53 $cm^{-1}$ due to the coincidence of the D and G peaks of GO and CNTs. Ordinarily, the degree of the defects of the composites is characterized by the ratio of the D peak and G peak ($I_D/I_G$) [36–38]. In comparison with the $I_D/I_G$ of GO and $CeO_2/CNTs$ (0.94, 0.99), the $I_D/I_G$ of $CeO_2/GO/CNTs$-d nanocomposite (1.05) increased. These results implied the changes in the structure of surface functional groups of nanocomposites and the formation of the $CeO_2/GO/CNTs$ nanocomposite. Specific results can also be observed in the XPS survey spectrum (Figure 4).

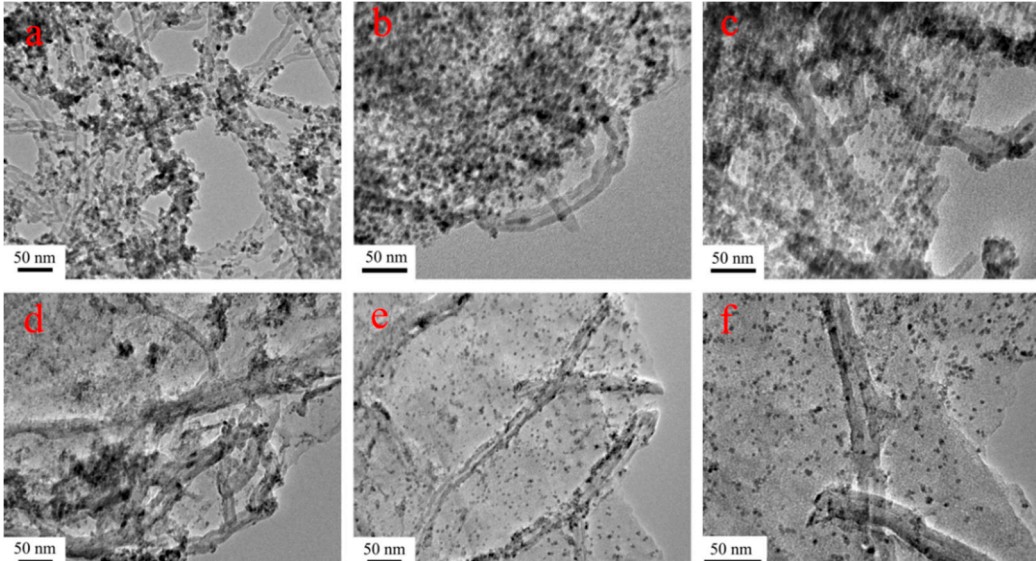

**Figure 2.** TEM images of (**a**) $CeO_2$/CNTs nanocomposites; (**b**) $CeO_2$/GO/CNTs-a nanocomposites; (**c**) $CeO_2$/GO/CNTs-b nanocomposites; (**d**) $CeO_2$/GO/CNTs-c nanocomposites; (**e**) $CeO_2$/GO/CNTs-d nanocomposites; and (**f**) $CeO_2$/GO/CNTs-e nanocomposites.

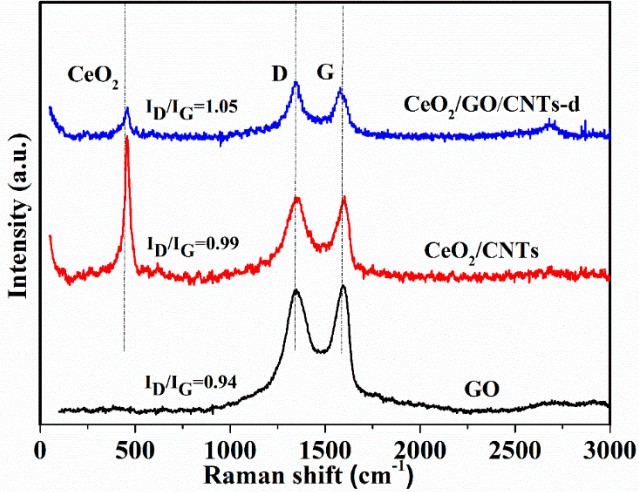

**Figure 3.** Raman spectra of GO, $CeO_2$/CNTs and $CeO_2$/GO/CNTs-d nanocomposites.

The elemental composition and chemical state of nanocomposites can be analyzed using XPS. The C1s XPS spectrum of $CeO_2$/CNTs nanocomposites shown in Figure 4a was deconvoluted into four peaks located at 289.2 eV (O=C–O), 286.5 eV (C=O), 285.6 eV (C–OH) and 284.8 eV (C–C), which were similar to the C1s XPS spectrum of $CeO_2$/GO/CNTs nanocomposites in Figure 4b [39]. From Figure 4c, the presence of the Ce–O bond can demonstrate the successful preparation of $CeO_2$/GO/CNTs nanocomposites via hydrothermal reaction. In Figure 4d, six Ce 3d peaks located at 917.4, 908.4, 901.5, 899.1, 886.3, and 882.9 eV, respectively, were assigned to $3d_{3/2}$ and $3d_{5/2}$ for $Ce^{4+}$ final states. Moreover, two weak peaks located at 902.25 and 889.7 eV should be assigned to $3d_{3/2}$ and $3d_{5/2}$ for $Ce^{3+}$ final states [33]. In combination with Figure 4b–d, it can be speculated that surfaces of both GO and CNTs were rich in oxygen-containing groups so that $CeO_2$ can be homogeneously compounded with them using a hydrothermal method.

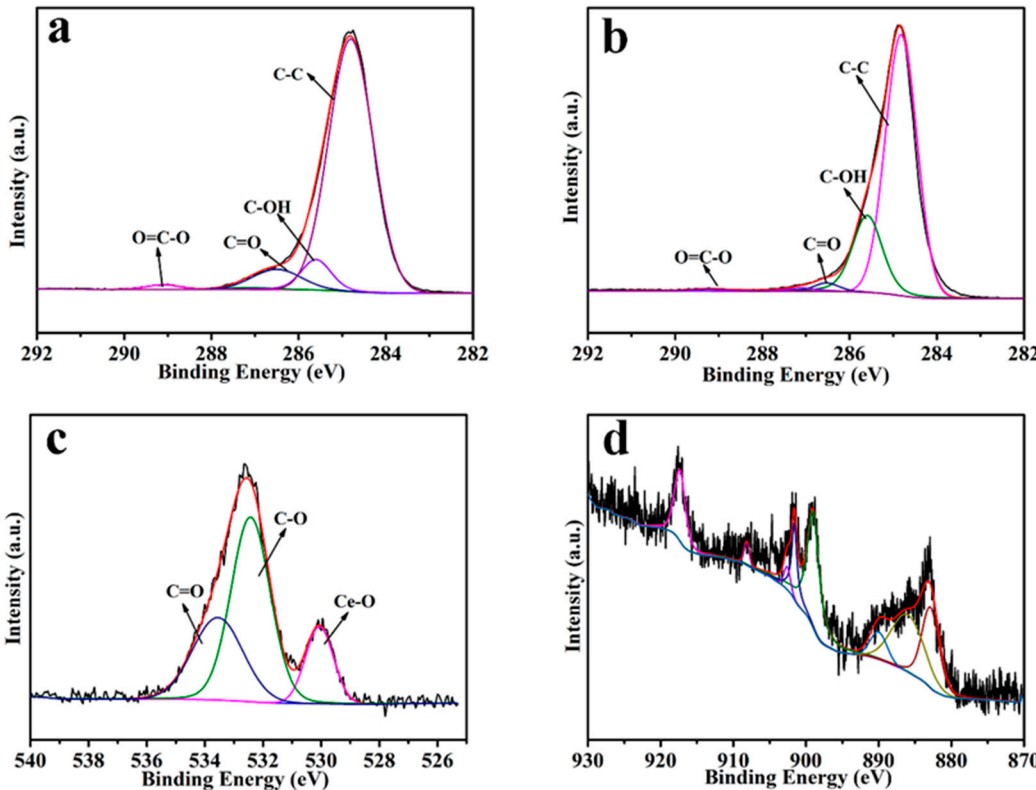

**Figure 4.** XPS survey spectrum for (**a**) C1s of $CeO_2$/CNTs nanocomposites; (**b**) C1s; (**c**) O1s; and (**d**) Ce3d spectrum for $CeO_2$/GO/CNTs-d nanocomposites.

The dispersion stability of lubricating additive is a significant factor that affects the tribological performance of lubricating oil. Paraffin oil used as the lubricant base oil in this study has a similar structure with petroleum and is more environmentally friendly than petroleum [40]. Figure 5 showed that 1 wt% $CeO_2$/GO/CNTs-d nanocomposites were dispersed in paraffin oil under ultrasonic agitation and placed in a water bath at 30 °C for observation. With the passage of time, the dispersion stability of nanocomposites in base oil was observed using a digital camera. We could clearly see that $CeO_2$/GO/CNTs-d nanocomposites possessed excellent dispersion and remained stable in the base oil on the twelfth day. The long-term dispersion stability of the nanocomposites in the base oil resulted from the uniform dispersion of $CeO_2$ nanoparticles on the GO/CNTs hybrids and the π-π bond interaction and rich oxygen-containing groups on the surface of nanocomposites, which could prevent the agglomeration between nanocomposites [30,31,41].

Generally, the friction performances of various nanomaterials used as additives and paraffin oil were characterized by the friction coefficient and wear scar diameters (WSD). Demonstrated in Figure 6 are variation of friction coefficient, average friction coefficient, and WSD of paraffin oil, 1 wt% of GO nanosheets, $CeO_2$ nanoparticles, $CeO_2$/CNTs nanocomposites, and $CeO_2$/GO/CNTs-d nanocomposites. Figure 6a shows that the paraffin oils filled with additives have superior stability compared with pure paraffin oil whose friction coefficient was increased with the passage of time. Additionally, the friction coefficient of paraffin oils filled with additives was lower than that of pure paraffin oil. It indicated that the addition of these additives can enhance the tribological performance of pure paraffin oil. Evidently, paraffin oils filled with $CeO_2$/GO/CNTs-d nanocomposites manifested the lowest friction coefficient. These results are consistent with average friction coefficient of abovementioned pure paraffin oil and paraffin oils filled with different additives shown in Figure 6b. WSD is of great significance for demonstrating the friction performance of paraffin oils. The trend of WSD of nanocomposites in Figure 6b is the same as the friction coefficient trend wherein the friction coefficient and WSD of paraffin oil filled with $CeO_2$/GO/CNTs-d nanocomposites decreased by 31.6%

and 37% compared with pure paraffin oil, respectively. The friction coefficient and WSD of paraffin oil filled with $CeO_2/CNTs$ nanocomposites did not follow the improvement trend. The aggregation of $CeO_2$ nanoparticles on CNTs was observed in Figure 2 and the $CeO_2/CNTs$ additives were prone to precipitate because of its poor compatibility with the lubricant base oil, which led to slightly influencing the improvement of tribological properties. Therefore, all these additives were effective for enhancing the tribological property of paraffin oil. In particular, the paraffin oil filled with $CeO_2/GO/CNTs$-d nanocomposites displayed excellent tribological properties. This can be explained by the fact that the two-dimensional lamellar structure of GO, the one-dimensional tubular CNTs, and the zero-dimensional spherical $CeO_2$ were uniformly compounded and achieved the synergistic lubrication effect [42] that the improvement of tribological properties of $CeO_2/GO/CNTs$-d nanocomposites was far superior to the single-component nanomaterials.

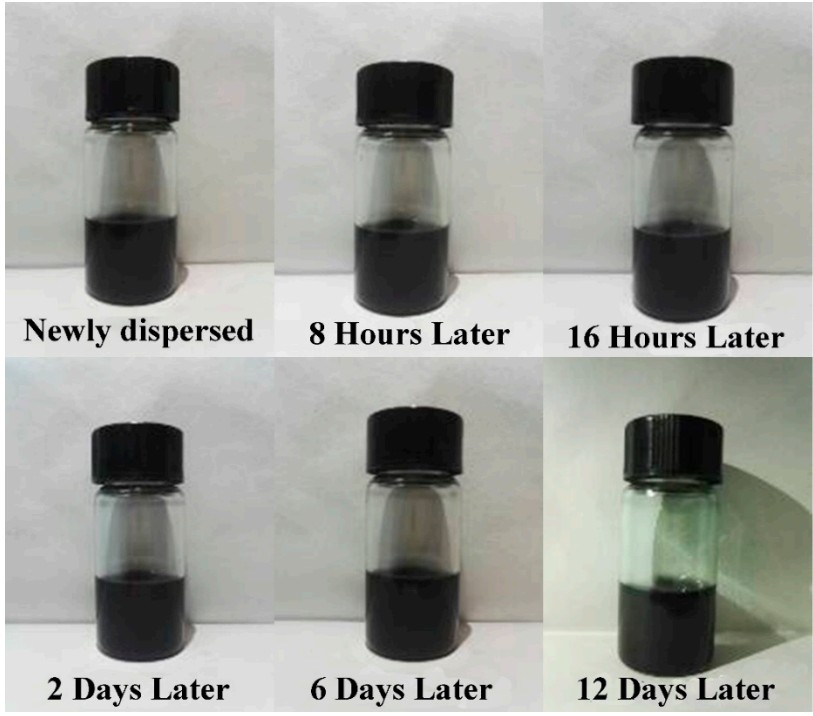

**Figure 5.** Digital photos of dispersion stability of 1 wt% $CeO_2/GO/CNTs$-d nanocomposites in paraffin oil at 30 °C.

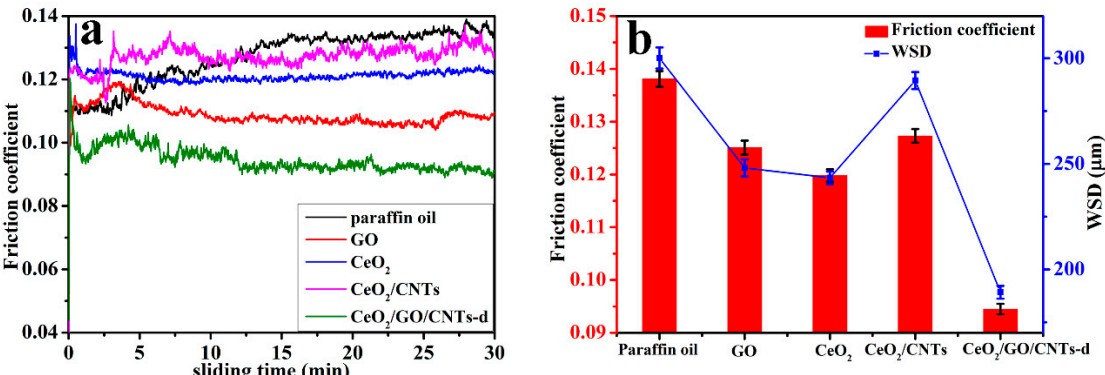

**Figure 6.** (**a**) Typical friction coefficient curves using the base oil and the oil with 1 wt% GO nanosheets, $CeO_2$ nanoparticles, $CeO_2/CNTs$ nanocomposites, and $CeO_2/GO/CNTs$-d nanocomposites; and (**b**) comparisons of average friction coefficient and average WSD of steel disks using these oils over the entire duration of the test (load of 10 N, rotating speed of 300 rpm, test duration of 30 min).

Considering the effect of different weight ratios of $CeO_2$, GO, and CNTs in the nanocomposite on the friction properties of paraffin oil, the tribological behaviors of $CeO_2$/GO/CNTs nanocomposites with a content of 1 wt% in paraffin oil were investigated as lubricant additives in pure paraffin oil. In Figure 7a, the paraffin oil filled with $CeO_2$/GO/CNTs nanocomposites using different weight ratios of raw materials gave different tribological properties. Therein, the friction coefficients of paraffin oil filled with the $CeO_2$/GO/CNTs-a and $CeO_2$/GO/CNTs-e nanocomposites presented large fluctuations and tended to rise slowly. In contrast to the instability of friction coefficients of paraffin oil filled with $CeO_2$/GO/CNTs-a and $CeO_2$/GO/CNTs-e nanocomposites, paraffin oil filled with $CeO_2$/GO/CNTs-(b–d) nanocomposites not only expressed the better stability but also had the lower friction coefficients. From the curve of friction coefficient and WSD shown in Figure 7b, it can be observed that $CeO_2$/GO/CNTs-d nanocomposites had the lowest friction coefficient and the narrowest WSD compared with other additives. The comprehensive analysis signified that GO/CNTs hybrids decorated with uniform $CeO_2$ nanoparticles were achieved at a proper mass ratio of GO/CNTs and $Ce(NO_3)_3 \cdot 6H_2O$, which is an important point to reduce the wear rate of base oil. Therefore, $CeO_2$/GO/CNTs-d possessed the superior lubricating performance owing to the uniform distribution of $CeO_2$ nanoparticles on surface of hybrids and no obvious aggregation of among $CeO_2$, CNTs, and GO, which is consist with the TEM images shown in Figure 2.

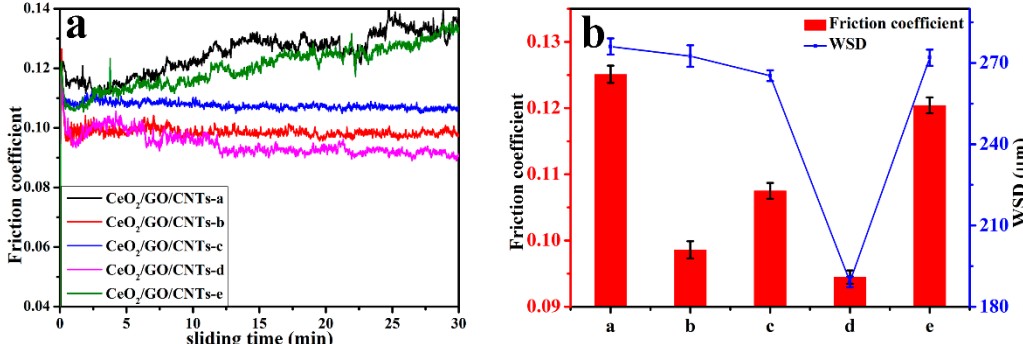

**Figure 7.** (**a**) Typical friction coefficient curves using the base oil with different nanocomposites; and (**b**) comparisons of average friction coefficient and average WSD of steel disks using these oils over the entire duration of the test (load of 10 N, rotating speed of 300 rpm, test duration of 30 min).

In order to obtain the appropriate content of $CeO_2$/GO/CNTs-d nanocomposites as a lubricant additive that improves the friction resistance of paraffin oil, the tribological performance of paraffin oil with different contents of $CeO_2$/GO/CNTs-d nanocomposites were investigated, as shown in Figure 8a,b. From the curve of friction coefficient with time in Figure 8a, it can be found that the $CeO_2$/GO/CNTs-d nanocomposites lubricant with a content of 0.5 wt% and 1 wt% were relatively stable. In particular, in Figure 7b, $CeO_2$/GO/CNTs-d nanocomposites with a content of 1 wt% manifested the lowest friction coefficient and the smallest WSD compared with other concentrations of additives. Initially, the friction coefficient and WSD declined with the increasing concentration of $CeO_2$/GO/CNTs-d. Base oil with low concentration (0.5 wt%) additives exhibited a higher friction coefficient and WSD due to the incomplete dispersion of additives in oil. While the concentration of $CeO_2$/GO/CNTs-d exceeded 1 wt%, the friction coefficient rose with the increasing concentrations of $CeO_2$/GO/CNTs-d, and the value of WSD was much more than 1 wt% $CeO_2$/GO/CNTs-d due to agglomeration. Therefore, 1 wt% $CeO_2$/GO/CNTs-d nanocomposites were the most suitable concentration, which makes the corresponding paraffin oil possess excellent friction resistance. The excellent lubricating performance was attributed to the perfect combination of the laminated GO nanosheets and the ball-bearing $CeO_2$ and CNTs [41–43].

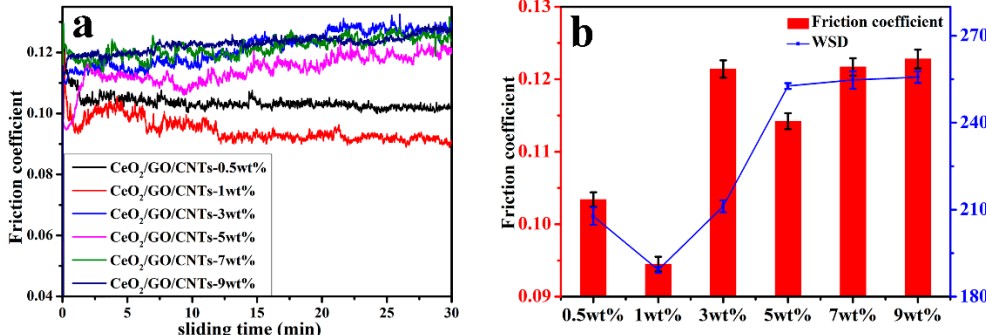

**Figure 8.** (**a**) Typical friction coefficient curves using the base oil with different contents of $CeO_2$/GO/CNTs-d nanocomposites; and (**b**) comparisons of average friction coefficient and average WSD of steel disks using these oils over the entire duration of the test (load of 10 N, rotating speed of 300 rpm, test duration of 30 min).

## 4. Conclusions

In this paper, 0-D $CeO_2$ nanoparticles, 1-D CNTs, and 2-D GO were used to formulate $CeO_2$/GO/CNTs nanocomposites using a hydrothermal reaction and the $CeO_2$ nanoparticles were uniformly distributed on GO and CNTs. The effect of $CeO_2$/GO/CNTs nanocomposites as a lubricating additive on the friction performance of paraffin oil were investigated. Compared with $CeO_2$/GO/CNTs-(a,b,c,e) nanocomposites containing the different weight ratios of $CeO_2$, GO, and CNTs, the paraffin oil with $CeO_2$/GO/CNTs-d nanocomposites was declared as having the lower friction coefficient and narrower WSD. Moreover, when the mass fraction was 1 wt%, as a lubricating oil additive, $CeO_2$/GO/CNTs-d manifested an excellent anti-friction performance. These results demonstrated that $CeO_2$/GO/CNTs nanocomposite additives possessed valuable properties for improving the tribological properties of lubricating oils due to the synergistic effect among three different size dimensions.

**Author Contributions:** C.M., H.S. and L.G. conceived and designed the experiments; W.J., J.Q., and Y.J. performed the experiments; D.L. analyzed the data; Z.H. wrote the paper.

**Funding:** This research was funded by the National Natural Science Foundation of China [51875330, 51803081, 51103065], the Tribology Science Fund of State Key Laboratory of Tribology [SKLTKF17B08] and the Project National United Engineering Laboratory for Advanced Bearing Tribology [201806]. And the APC was funded Postgraduate Research & Practice Innovation Program of Jiangsu Province [KYCX18_2233].

**Conflicts of Interest:** The authors declare no conflict of interest.

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
