# Peer review of "Fabrication of Novel CeO2/GO/CNTs Ternary Nanocomposites with Enhanced Tribological Performance"

_applsci, doi:10.3390/app9010170_

Reviewer 1 Report

applsci-414304: Fabrication of novel CeO2/GO/CNTs ternary 2 nanocomposites with enhanced tribological 3 performance, Chunying Min, Zengbao He, Dengdeng Liu, Wei Jia, Jiamin Qian, Yuhui Jin, Haojie Song, and Li Guo.

Technical comments (not in order of importance):
Raman intensity ratio is good index for the evaluation of carbon materials abundance. Takagi et al, (Journal of Nanoscience and Nanotechnology Vol.8, 2665–2670, 2008) used Raman technique to investigate structural change of CNTs before and after friction tests, and concluded that formation of CNT-derived transferred films lead to induce lower friction coefficient characteristics. I expect that Takagi’s work yields valuable information for strengthening this manuscript. The authors investigate the CNT and GO structures after friction tests in order to reveal possible mechanisms observed for CeO2/GO/CNT nanocomposites with better friction and wear properties.

Line 210: What kind of synergistic lubrication effect do the authors expect? It should be noted in the manuscript.

The possible reasons why better friction and wear properties were obtained in the CeO2/GO/CNT-d and the CeO2/GO/CNT with 1wt% should be clearly stated in the manuscript.

Author Response

Dear reviewer:

  We appreciate greatly you for your comments and suggestions on our manuscript ID applsci-414304.Those comments are all valuable and very helpful for revising and improving our paper, as well as the important guiding significance to our researches. We have carefully rechecked the manuscript based on the comments. In addition, the changes we made have been colored in red using the "Track Changes" function in our revised manuscript. The list of responses to the comments is attached.

Sincerely yours,

Chunying Min

Reviewer 2 Report

This paper concerns about the lubrication action of a nanocomposite based on CeO2/GO/CNTs system added to paraffin oil. The Author presented their results on the preparation and characterization of the nanocomposites with different weight ratio of the various cimponents. They characterized the materials in order to define the chemical composition, the degree of dispersion of Cerium particles and verify if this nanocomposite, added to paraffin oil, could have a better lubricating action than just paraffin oil.

The discussion of the article is fluent, well organized and presented. It is easy to follow and the results are pleasant. Anyway some points needs to be better discussed.

In detail, list the minor revisions sugegsted for this paper:

1       Introduction

-This part is a little weak from the point of view of the innovative nature of the material. So I ask the authors to specify this aspect and if other Authors studied similar systems and the results obtained (comparing them);

-in line 77 the Authors states that CNT and GO are combined through a π- π bond. Where , in the text , is discussed this experimental evidence?

2 Experimental

pag 3/11, line 100-104: please specify better codes of the samples ( for example by adding the word “respectively” after each group of nanocomposites.

3 Results

-        Pag.4/11 line 139-140: how do the Authors explain the fact that the CeO2 / GO / CNTs-d nanocomposite has the best dispersion and therefore the best performance from a lubricant point of view compared to the other combinations of weight ratio of the components?

-        Pag.6/11, line 178-179: The sentence “ Paraffion oil as the lubricant……petroleum” should be supported by letterature evidences as : J. Materials Science & Engineering C (2017)  http://dx.doi.org/10.1016/j.msec.2016.11.088

-        Pag.6/11, line 184-186: The sentence “ The high dispersibility ……nanocoomposite” should be supported by letterature evidences, as well.

-        Pag.8/11, line 211- fig.6b: how do the Authors explain the fact that the CeO2/CNTs does not follow the improvement trend of this graph? ie why the combination of cerium oxide with CNTs does not give a better lubricating action than cerium oxide alone? add a comment, please.

-        Pag.9/11, line 246-248: Is the sentence “ …lamellar structure of GO …friction.”  supported by other letterature evidences and hence, by other experimental evidences?

Author Response

Dear reviewer:

We appreciate greatly you for your comments and suggestions on our manuscript ID applsci-414304.Those comments are all valuable and very helpful for revising and improving our paper, as well as the important guiding significance to our researches. We have carefully rechecked the manuscript based on the comments. In addition, the changes we made have been colored in red using the "Track Changes" function in our revised manuscript. The list of responses to the comments is attached.

Sincerely yours,

Chunying Min

Round  2

Reviewer 1 Report

Necessary amendments have been made in the revised manuscript.